# High-resolution modeling and projection of heat-related mortality in Germany under climate change
Junyu Wang [1] ✉, Nikolaos Nikolaou [2], Matthias an der Heiden [3] & Christopher Irrgang [1]

## Abstract

**Background** Heat has become a leading cause of preventable deaths during summer. Understanding the link between high temperatures and excess mortality is crucial for designing effective prevention and adaptation plans. Yet, data analyses are challenging due to often fragmented data archives over different agglomeration levels.

**Method** Using Germany as a case study, we develop a multi-scale machine learning model to estimate heat-related mortality with variable temporal and spatial resolution. This approach allows us to estimate heat-related mortality at different scales, such as regional heat risk during a specific heatwave, annual and nationwide heat risk, or future heat risk under climate change scenarios.

**Results** We estimate a total of 48,000 heat-related deaths in Germany during the last decade (2014–2023), and the majority of heat-related deaths occur during specific heatwave events. Aggregating our results over larger regions, we reach good agreement with previously published reports from Robert Koch Institute (RKI). In 2023, the heatwave of July 7–14 contributes approximately 1100 cases (28%) to a total of approximately 3900 heat-related deaths for the whole year. Combining our model with shared socio-economic pathways (SSPs) of future climate change provides evidence that heat-related mortality in Germany could further increase by a factor of 2.5 (SSP245) to 9 (SSP370) without adaptation to extreme heat under static sociodemographic developments assumptions.

**Conclusions** Our approach is a valuable tool for climate-driven public health strategies, aiding in the identification of local risks during heatwaves and long-term resilience planning.

## Plain Language Summary

Heat is becoming a major cause of preventable deaths during the summer. We developed a computer model to estimate heat-related deaths at specific times and in different districts. Using this model for Germany, we estimate that over the past decade (2014–2023), around 48,000 deaths were heat-related, with most occurring during heatwaves. For example, a heatwave from July 7–14, 2023, contributed to 1100 out of 3900 heat-related deaths that year. Our model also suggests that, without adaptation, heat-related deaths in Germany could increase remarkably due to climate change. The insights from our model can help identify areas at high risk and support long-term public health planning to reduce the impact of heatwaves.

Human civilization has become the driving force of the Earth's changing climate, as anthropogenic greenhouse gas emissions are the primary cause of the recent global warming[1]. Since the industrial revolution, referenced to the time period of 1850–1900, the average global surface temperature has risen by about 1.2 °C[2]. Europe, for instance, is warming at almost twice the global rate, with an average temperature increase of around 2.2 °C[2].

Rising global temperatures lead to a cascade of global and regional phenomena and impacts, such as ice sheet loss and sea level rise[3], ecosystem disruption[4], and extreme weather events that increase in frequency, duration and severity[5,6]. Heatwaves, prolonged time periods of excessive heat over a given spatio-temporal extent, are one type of extreme events that have been linked to anthropogenic climate change repeatedly. Over the past decades, heatwaves have increased globally and regionally[7], posing direct threats for human health[8], public health systems[9], food security[10], work productivity[11], and infrastructure[12]. Higher temperatures are strongly linked to an increase in heart- and lung-related health issues, ultimately resulting in higher mortality rates[13]. The overall burden of heat-related mortality has risen on all continents[14–17] with particular risks for ageing sub-populations[14]. In cities and larger urban agglomerations, the so-called heat island effect can further amplify ambient temperatures by up to 10°C[18], causing additional heat stress on the human body. In 2022, it was estimated that 60,000 people in Europe died directly due to high temperature[19]. In Germany, the unusually high-temperature summers of 2018–2020 have been estimated to attribute to almost 20,000 heat-related deaths[17].

[1]Centre for Artificial Intelligence in Public Health Research, Robert Koch Institute, Berlin, Germany. [2]Institute of Epidemiology, Helmholtz Munich, German Research Center for Environmental Health, Neuherberg, Germany. [3]Department of Infectious Disease Epidemiology, Robert Koch Institute, Berlin, Germany. ✉e-mail: WangJ@rki.de

As a consequence of the rising heat-related burden, temperature has become an essential global monitoring variable to assess current and future health risks[20]. Currently, we are on track for 2.5–2.9 °C of global warming under current climate pledges this century[21], which will further intensify the described climate impacts and, thus, also the burden on our health systems. Several nations are in the process of shaping and implementing adaptation and safety plans against extreme heat[22]. Although many studies have assessed the impact of heat on mortality, most either provide risk estimation for large regions or only focus on cities. Combined risk estimation and projection across different spatial and temporal scales is currently still missing, making it challenging for local governments to develop targeted preventive and adaptive plans. Accurate and high-resolution estimates of the heat-related stress for public health systems are crucial for such purpose.

The connection between temperature and mortality is often examined with generalized additive models (GAMs), distributed lag models (DLMs), and similar statistical models[17,19,23], despite their limitations. In most models, the effects of temperature on each day are considered independent[24]. However, consecutive hot days may result in additional heat stress, and such cumulative effects are not captured by the models. Also, the model calibration depends on both temperature and mortality data, but these data sources often do not align in terms of their time and location specificity. For example, temperature data can be available on a daily or even sub-daily basis with kilometer-scale resolution. In contrast, mortality data is often reported weekly at state or nation level. This discrepancy in temporal and spatial resolution usually leads to the use of aggregated temperature products over a larger area as predictors for estimating mortality, which can degrade the model's precision. Recently, Ballester et al. studied the impact of temporal data aggregation on the estimation of the excess death cases and showed that using weekly temperature as input could result in a 20% underestimation of heat-related excess mortality[25]. Additionally, most models overlook the impact of regional temperature variations, especially heat island amplification, which can increase heat-related deaths during heatwaves[26].

In this study, we developed a multi-scale machine learning model to estimate daily heat-related mortality in Germany at the district level. The term 'multi-scale' refers to the utilization of data with varying temporal resolutions, ranging from daily to weekly, and data with different spatial resolutions, from $1 \times 1$ km grids to statewide. This multi-scale approach enables a risk assessment with higher temporal and spatial resolutions comparing to other models. By analyzing large-scale heat risk, including annual and nationwide estimates, as well as regional heat risk, such as urban agglomerations during specific heatwaves, our model provides comprehensive insights. Furthermore, we combined our model with climate projections from different shared socioeconomic pathways[27] to explore the impact of climate change on the heat-related mortality in Germany up to the year 2100. This analysis provides a detailed multi-scale heat-related burden and risk analysis for Germany, serving as a blueprint for other countries with similar data availability.

With our model, we find that approximately 48,000 heat-related deaths have occurred in Germany over the past decade (2014–2023), with most cases concentrated during specific heatwaves. Our projections indicate that without adaptation, heat-related mortality could increase by a factor of 2.5 to 9 by 2100 due to climate change. These findings highlight the urgent need for targeted adaptation measures to reduce future heat-related health risks.

## Methods
### Data sources
**Mortality.** We acquired all-cause mortality counts from the Federal Office of Statistics of Germany[28,29]. We used three statistics as the targets for our training process: Daily mortality by state and gender; weekly mortality by state, gender, and broad age groups (0–65, 65–75, 75–85, 85+); as well as weekly mortality data for Germany as a whole, categorized by gender and detailed age group (0–30, five-year age groups from 30–35 to 90–95, 95+). Ethics approval was not required for the use of this data because sufficient anonymization is achieved through aggregation. The data are aggregated at the district level, ensuring that individual identities cannot be traced or identified, in compliance with data protection regulations.

**Population.** We obtained population data information for each district (total of 400) as of December 31st annually from the regional database of Germany (population estimate[30]). We selected data from the year 2011 onward because the counting methods changed that year. The population of each district was categorized by gender and age group (five-year age groups from 0–5 to 90–95, 95+). We interpolated the population data to estimate daily population figures.

The projection of demographic changes until 2070 is available from the German Federal Statistical Office[31]. To estimate the district-level population, we assumed that the geographical distribution of each age group remains the same as in 2021.

**Temperature.** We utilized three sources to collect past and present temperature data for Germany. First, we acquired gridded temperature data from the Copernicus European Regional ReAnalysis (CERRA), covering the years 2011 to 2020[32]. The data were used to compute 6-hourly average temperatures for each district in Germany. Second, we used daily minimum, average, and maximum temperatures at a spatial resolution of $1 \times 1$ kilometer provided by Helmholtz Munich[33]. This dataset was obtained under a Data Transfer Agreement between our research institution and Helmholtz Munich, ensuring secure and authorized access to the data. We noticed some missing values in the dataset from Helmholtz Munich and supplemented it with data from the CERRA dataset while considering the mean difference between the two datasets. We also noticed the irregularity of the mean temperature in year 2013 from Helmholtz Munich, which are much lower than other years. However, we retained the data as-is for training purposes in this paper, but excluded it from metric evaluation. Data beyond 2020 was excluded from training and validation due to the impact of COVID-19 on mortality.

The above mentioned datasets are available on a yearly basis and are not suitable for real-time estimation of the heat-related risk. To address this issue, we utilized temperature station data from the Deutscher Wetterdienst (DWD)[34]. The station data are available on a daily basis. We gathered information from 537 weather stations. These stations have been measuring the average daily air temperature 2 meters above the ground since 2010 and have continued until at least 2022. The location of the stations is illustrated in Supplementary Fig. 1. In order to unify the spatial coverage in accordance with the reanalysis data, we trained an attention model to map the daily average temperature from the DWD stations to the district-level average temperature.

**Geodata.** The geographic information including the boundary and reference point of each district in Germany was provided by Federal Agency for Cartography and Geodesy (BKG). The boundary of each district was used to calculate the temperature in each district. The reference points were used to initiate the coordinate of each district during the training of the temperature attention model.

**Climate projections.** We obtained climate projections from the EC-Earth3 model for three different shared socio-economic pathways (SSPs) as defined by the Intergovernmental Panel on Climate Change (IPCC): SSP126, SSP245, and SSP370[35]. The dataset includes daily mean temperatures spanning from 2015 to 2100, with a spatial resolution of 100 km. The projection data of each scenario is a climate ensemble of 50 members. To map the temperature from the grid to the district level, we obtained daily average temperature with the ERA5 daily statistics calculator provided by the Copernicus Climate Data Store[36] with a grid of $0.5° \times 0.5°$ for training a down-scaling model. We interpolated the ERA5 data to match the grid of the EC-Earth3 model and used the interpolated values to train a machine learning model for climate downscaling.

## Machine learning model for mortality predictions

Our objective is to predict the number of heat-related excess death cases in each district for each age group and sex. The estimation of all-cause mortality operates as follows:

$$E_{\text{district,age,sex}}(\text{mort}) = \text{baseline}_{\text{age,sex}} \times \text{population}_{\text{district,age,sex}} \times f_{\text{age,sex}}(T_t). \tag{1}$$

We used temperature data ($T_t$) of each district at time $t$ as input for our model to obtain the result $f_{\text{age, sex}}(T_t)$. The function $f$ is used to describe the temperature dependent variability of temperature-related risks. The model accounted for the lag effects of temperature by using different convolutional kernel size. By multiplying $f_{\text{age, sex}}(T_t)$ with the baseline death rate and population data, we obtained the final results, comprising daily death cases for each district, categorized by gender and age groups. These results were subsequently aggregated and compared to the registered death cases to calculate the model's loss, which was used to train the model.

More specifically, we predicted multiplying factors for the baseline mortality rate using temperature as input. We assumed a uniform baseline mortality rate for each gender and age group across all districts within Germany, with the baseline depending solely on the time. Multiplying these factors with the baseline provided the predicted daily mortality rates for each district. Unlike conventional statistical models, we did not explicitly account for seasonal trends, as these were implicitly encapsulated within the temperature variable. Along with the population data for each district, we predicted the daily death cases for each district according to genders and age groups. We aggregated the predicted daily death cases to align with the temporal and spatial scope of the data from the Federal Office of Statistics of Germany. We employed the Poisson loss function to calculate the loss between the aggregated prediction and the registered death cases and utilize this loss to train our model. If correction factors were applied for different days of the week, the correction was performed before aggregating the predicted death cases.

Throughout our analysis, we consistently partitioned the data into training and validation sets, maintaining an 80:20 ratio with data from 2011 to 2018 used as training data and data from 2019–2020 as validation data. We evaluated two distinct network architectures to fulfill our research objectives.

**Linear model**. In the linear model, we employed a fully connected network with ReLU (rectified linear unit) as the activation function. To investigate the temperature-lag-mortality relationship, we incorporated a 1-dimensional convolutional layer with kernel size $l$ as the initial layer within the neural network architecture to capture the lag effect of the temperature on the mortality.

**Exponential model**. The exponential model employed a shallow network to extend the statistical model and permitted the summation of exponential terms, a feature absent in GAMs and DLMs. The structural principle of the network can be described by the following function:

$$f(T) = g\left(\exp(g_1(T)), \ldots, \exp(g_d(T)), \quad T = [T_0, T_1, \ldots, T_l]. \tag{2}\right)$$

Here, $T$ is the input temperature vector, $l$ represents the number of lag days under consideration, and $d$ denotes the network's hidden dimension. The function $g, g_1, \cdots, g_d$ embodies the affine transformations executed by the network. The scaling factor within the function $g$ is non-negative. The resulting output, denoted as $f$, is the multiplying factor of the baseline and is then used to calculate the predicted death cases for each district, classified by gender and age group.

The performance of a single trained model depends strongly on the initial value. There can be significant differences among models trained with different initial values. Therefore, we trained multiple instances of the temperature-mortality models and used the average output of the ensembles as the final result to reduce prediction variance.

## Machine learning model for district level temperature estimation

We used an attention-like model to interpolate the data from the DWD stations or climate projections. After this interpolation step, we obtained the data that serves as the input for our temperature-mortality model. In the attention-like model, each district was assigned a trainable geographical position $X_i = (\text{lon}_i, \text{lat}_i)$, where $i$ identifies different districts. Similarly, a geographical position $Y_j = (\text{lon}_j, \text{lat}_j)$ was assigned to each DWD station or grid point in the climate projections according to their location. The daily average temperature for a district, denoted as $T_i$, was calculated as follows

$$T_i = b_i + \sum_j w_{ij} T_j, \text{ where } w_{ij} = \frac{e^{-\alpha_i d(X_i, Y_j)}}{\sum_k e^{-\alpha_i d(X_i, Y_k)}}. \tag{3}$$

Here, $b_i$ is the trainable bias, $T_j$ is the temperature at the corresponding DWD station or grid point in climate projections, $w_{ij}$ is the attention of district $i$ on station or point $j$, and $d$ calculate the distance between $X_i$ and $Y_j$, and $\alpha_i$ is a trainable district specific scaling factor.

The temperatures acquired this way capture the local weather exactly (RMSE compared to the Helmholtz Munich dataset, 0.23 °C for training dataset, 0.25 °C for validation dataset) and include the impact of the heat island effect implicitly. The district level temperatures can then be used as input for our model to provide accurate mortality estimation.

### Statistics and Reproducibility

To assess the stability of the machine learning models, multiple instances of each setup were trained with different random initial values. The number of instances for each model setup is listed in the corresponding Supplementary Tables 2–7. Performance was characterized using mean square errors and coefficients of determination ($R^2$-values). For the final setup, 20 instances were trained, and the average of these 20 instances is reported as the final result.

### Reporting summary

Further information on research design is available in the Nature Portfolio Reporting Summary linked to this article.

## Results
### Past and present heat-related mortality in Germany

Figure 1 shows the relationship between the aggregated number of warm days (>20 °C) in each district and the estimated heat-related annual mortality risk for the past six years (2018–2023). We used a temperature cap of 20 °C as a constraint of baseline mortality in the absence of heatwaves to be consistent with previous publication from RKI[17]. (The impact of the cap temperature is presented in Supplementary Fig. 2). The differences between the estimations with original and capped temperatures represent the estimated mortality risk caused by high temperatures[17]. While the year-to-year spatial extent of past high temperatures shows a complex pattern on the district level, certain hot-spot regions become visible. The areas most heavily impacted by the high temperatures include the western region of North Rhine Westphalia (Ruhr area), Saarland, the northwest of Baden-Württemberg, Berlin, Brandenburg, Saxony. Due to the high spatial resolution of the temperature data, also the heat island effect becomes evident. This effect is visually represented by the small red areas (district-free cities and other large urban agglomeration areas, e.g., Berlin and the Ruhr area) on the maps. The cities had more warm days compared to the surrounding areas, resulting in more heat-related deaths. Based on our model estimates, regions with 45 or more warm days in a year are likely to experience a heat-related mortality rate of 100 per million population per year or more.

Additional to the annual estimation of heat-related mortality, our model enables a near real-time (daily) heat-related risk estimation at the district level. This combination allows further insights into the impact of individual heatwaves and their contribution to the previously estimated annual heat-related mortality. As an example, we selected the week of July 7–14, where most of Europe was exposed to extreme heat[37]. Figure 2

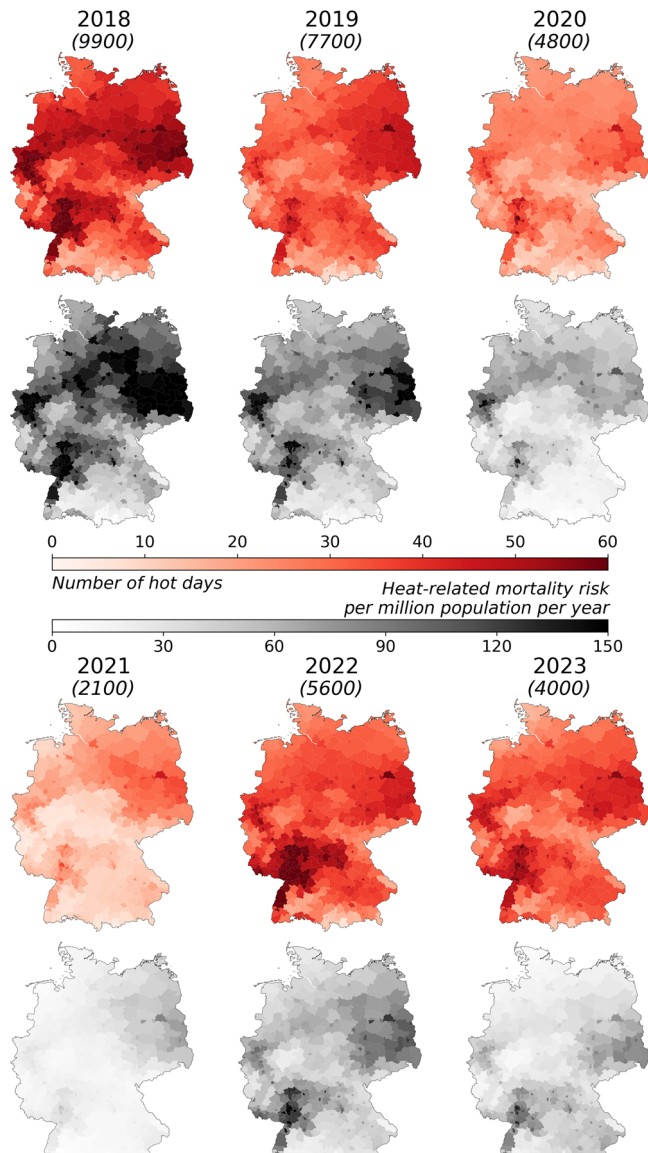

**Fig. 1 | Yearly aggregated number of hot days and heat-related mortality risk between 2018 and 2023.** Numbers in parentheses indicate the estimated annual heat-related mortality. Red maps: Number of days with a daily average temperature over 20 ˚C at the district level (2018-2020: Temperature data from Helmholtz Munich; 2021-2023: Interpolated DWD station data). Grayscale maps: Estimated yearly heat-related mortality rate per million people at the district level (best model with day-of-the-week correction, based on 20 ensemble members). Due to COVID-19, a reliable baseline for mortality rates couldn't be calculated for 2021–2023, so the baseline mortality from the last week of 2020 was used instead. Our model's estimated heat-related deaths closely matched those in the RKI reports when data with similar spatial resolution was used as input (see Supplementary Table 1).

illustrates the dynamics of estimated mortality risks between July 7 and 14, with a 5-day heatwave between July 8 and 12. The estimated mortality risks in all districts were less than 1 per million people per day on 7 July, one day before the heatwave. On the first day of the heatwave, 8 July, the estimated mortality risks increased slightly in the west and southwest of Germany, where temperatures were also higher. Temperatures increased further on 9 July, with a nation-wide average of 24.5 °C, the peak during the heatwave. The estimated mortality risks also increased, but did not peak until three days later, on 12 July, even though the weather cooled down slightly (21.1 °C) compared to 9 July (24.5 °C) and 11 July (23.9 °C). The heatwave ended on 12 July, but the elevated mortality risks persisted for another day

(13 July) before fading away on 14 July. We estimated a total of 1100 heat-related deaths between 7 and 14 July. All federal states, except for Schleswig-Holstein, Hamburg, Lower Saxony, Bremen and Mecklenburg-Vorpommern in the north, experienced a mortality risk of over 10 per million people during the heatwave. Peak values were estimated in Rhineland-Palatinate with a risk up to 39 deaths per million people. This analysis indicates a distinct lagged relationship between excess heat and daily mortality risk that persists beyond the temperature peak for up to two days, highlighting the need for preventive and adaptive measures even when temperatures start decreasing after the peak of a heatwave. Compared to the estimated total heat-related mortality in the summer of 2023 (Fig. 1), this single event contributed approximately 28%.

### Scenario-based future heat-related mortality in Germany

The learned exposure-response function in our model allows us to investigate future heat-related mortality risk under climate change. We applied our model to process different climate projections, provided through socio-economic pathways (SSP) as defined by the sixth assessment report of the Intergovernmental Panel on Climate Change (IPCC, [38]). We utilized data from the sustainability scenario (SSP126, estimated global warming of 1.8 °C until 2100), the middle of the road scenario (SSP245, estimated global warming of 2.7 °C until 2100), and the regional rivalry scenario (SSP370, estimated global warming of 3.6 °C until 2100). In line with the past and present temperature data used above, we applied an attention mechanism to derive daily temperature projections at the district level and used them as input for the machine learning model (see details in the methods section). We further assumed that population and death rates would remain constant to simplify mortality projections.

The SSP-based temperature trajectories and associated heat-related excess mortality estimate are illustrated in Fig. 3. Excess mortality is expected to respond non-linearly to the increasing temperatures, as both average and extreme temperatures (heatwave occurrence and frequency) are rising, especially in the SSP370 scenario. Over the coming decades until 2050, the excess mortality shows comparable increments, particularly in the SSP126 and SSP245 scenarios. It is noteworthy that SSP245 assumes current $CO_2$ emissions until 2050. Between 2050 and 2100, the excess mortality in SSP126 remains almost constant, while the excess mortality in SSP245 and SSP370 shows distinct increase. By the last decade of the 21st century, annual median excess deaths of approximately 3700 (SSP126), 11,600 (SSP245) and 41,000 (SSP370) could be attributable to heat in Germany, which amount to 40, 140, and 500 per million people per year, respectively.

As a comparison, we included two non-static demographic scenarios extending until 2070 to assess the additional demographic impact on the future projections of heat-related health risks (see Supplementary Fig. 3 and discussion below).

### Accurate prediction of summer mortality

Many studies have shown that there is a minimum mortality temperature, and the mortality rate increases monotonically for temperatures above and below this optimum[39]. Using temperature as the sole input, our model successfully captures the seasonal trends in mortality (Supplementary Fig. 4), with higher mortality in winter and lower mortality in summer. The model's performance varies with different setups; the results are provided in Supplementary Fig. 5 and Supplementary Tables 2–6, with further discussion in Supplementary Discussion.

Supplementary Table 7 demonstrates that our model had an accurate estimation of the daily mortality cases in Germany for warmer days (nationwide mean temperature above 20 °C. Root-mean-square Error (RMSE): 91.4 for training data, 83.9 for validation data. Coefficient of determination ($R^2$): 0.7887 for training data, 0.8272 for validation data.). For comparison, the theoretical lower bound of the RMSE falls within the range of 45 to 60 according to the Poisson distribution and the upper bound of the $R^2$ is around 0.94. However, when considering all days in a year, the predictions are less accurate (RMSE: 171.1 for training data, 199.6 for validation data. $R^2$: 0.5961 for training data, 0.4884 for validation data.). The decrease

**Fig. 2 | Daily temperature and heat-related mortality risk dynamics during the heatwave from July 7 to July 14, 2023.** The temperature below the date represents the nationwide average. Numbers in parentheses indicate the estimated daily heat-related mortality. Red maps: Daily average temperature at the district level, interpolated from DWD station data. Grayscale maps: Estimated daily heat-related mortality risk per million people at the district level (best model with day-of-the-week correction, based on 20 ensemble members). Due to the aforementioned reasons, the baseline mortality from the last week of 2020 was used.

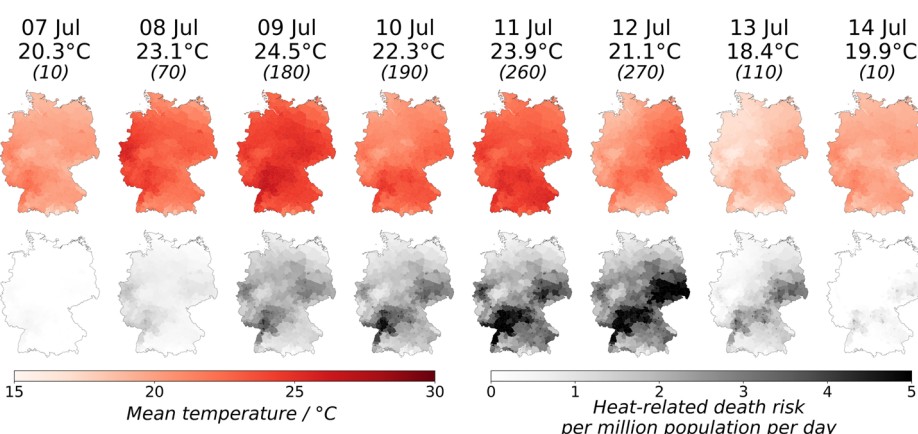

**Fig. 3 | Temperature projections and estimated heat-related excess mortality in Germany (2015–2100) under different climate change scenarios. a** Projected annual average temperature from the EC-Earth3 model. **b** Projected annual heat-related excess mortality cases (best model without day-of-the-week correction, based on 20 ensemble members). Excess mortality was calculated assuming constant population and baseline mortality rates as of the end of 2020. The lines indicate the median, and the shaded areas represent the 10th to 90th percentile range calculated from the climate ensembles, with 50 members for each scenario.

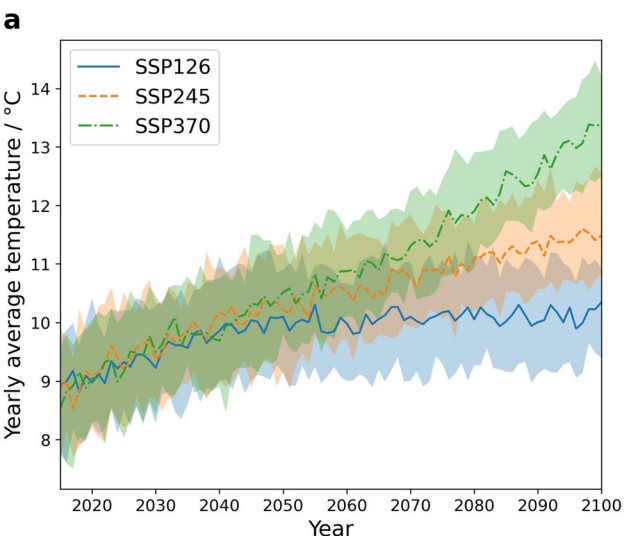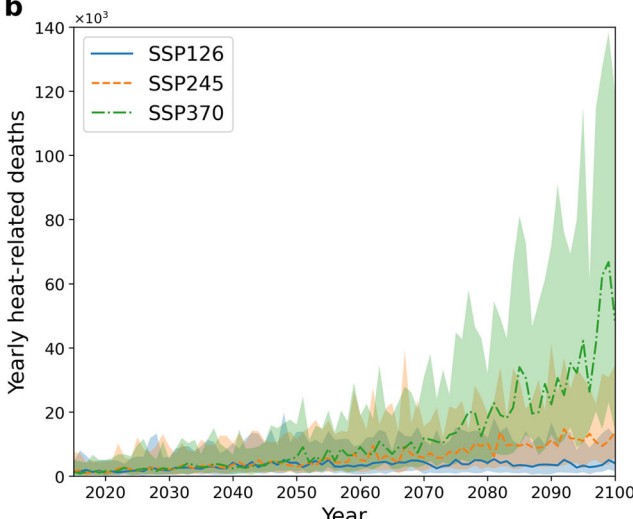

in performance is attributed to some notable deviations in the early months of 2015, 2017, and 2018. In those years, severe influenza waves occurred in Germany[40]. Another notable deviation occurred at the end of 2020, when COVID-19 surged worldwide. The registered death cases are higher than the prediction of the model, because our model only used temperature as input for mortality prediction and cannot include the impact of transmission diseases. These results highlight that in summer most excess deaths are attributed to heat, while in other seasons the mortality can be affected by additional factors, for example, seasonal influenza.

We compared our mortality predictions with the registered death cases and the predictions in the weekly reports from Robert Koch Institute (RKI)[41]. For weeks with an average temperature above 20 °C, our model demonstrated performance similar to that of the statistical model used in RKI reports (weekly mortality RMSE from 2011 to 2020, except for 2013: 362.1 for our model, 364.9 for the GAM model used in RKI reports, weekly $R^2$: 0.8997 for our model, 0.8982 for the GAM model used in RKI reports, see Supplementary Fig. 6).

## Discussion
We developed a multi-scale machine learning method that integrates population dynamics modelling with temperature data and mortality statistics to provide district-level heat-related mortality estimates for Germany.

The model addressed the scope and scale mismatch between climate and mortality data. In comparison to other models, this approach allowed us to account for daily weather variations at the district level and to demonstrate the spatial impact of urban heat islands. Overall, we estimated 48,000 heat-related deaths in Germany during the last decade (2014-2023), of which most cases can be attributed to individual heatwaves.

In addition to the estimation of past heat-related mortality, our model enables near real-time risk estimation at the district level. Assuming accurate documentation of the death date, the example of the heatwave between July 7 and 14 of 2023 clearly illustrates a temporal delay between the onset of a heatwave and the increase in health risk. Even when the heatwave ended, the elevated risk persisted. The lag effect of the heatwave was limited to 3 days. Consequently, this approach could be used in combination with weather forecast systems to assess the heat-related risk in the near-term future.

The contribution of heatwaves to the total heat-related mortality has not been extensively studied. Xu et al. compared different definitions of heatwaves in a review paper and showed that defining a heatwave across large areas is difficult due to the variations in population acclimatization and adaptation[42]. Pascal et al. suggested that 6% of days account for 28% of the heat-related mortality from 2014 to 2022 in France[43]. In cities in Latin America, the extreme heat contributed to approximately 60% of the total heat-related mortality[44]. In this paper, we did not explicitly define a

heatwave; instead, our model captures heatwave characteristics through temperature and mortality data. According to our model, the top 10 days with the highest heat-related deaths each year contributed to 40–90% of the total heat-related mortality of the corresponding year in Germany (see Supplementary Table 8). Only 7% of the days are disconnected, which demonstrates the impact of consecutive hot days, i.e., heatwaves. These findings also emphasize the importance of focusing heat risk prevention plans on adapting to heatwaves. Further research is essential to compare the temporal distribution of heat-related mortality within a year across different countries and to provide valuable insights for the development of effective heat-health prevention plans.

Climate projections based on the IPCC's shared socio-economic pathways indicate that heat-related mortality in Germany is likely to further increase without dedicated heat adaptation measures. In the middle of the road scenario (SSP245), which is considered most likely based on current policies, the annual heat-related excess mortality by 2100 could be 2.5 times higher than today, which would amount to over 10,000 heat-related deaths on average in a single year. In the higher emission scenario (SSP370), the median excess mortality in Germany could be 9 times higher than today. In the sustainability scenarios (SSP126), in which global warming is limited to below 2 °C, the heat-related mortality would be constrained to around 5000 annual losses on average, staying within the present heat-related mortality dynamics. These estimations did not account for the population development, for instance, expected population aging[45], which is considered as another risk amplification factor. We investigated this effect by utilizing two different non-static demographic projections until the year 2070, simulating a range of future population aging scenarios that are driven by varying birth rates, life expectancy, and net migration (Supplementary Fig. 3). Independent of the climate scenario, we observe a steadily increasing additional risk factor until at least 2050 in the range of 40 to 60% compared to the static population assumption. These results suggest that the expected demographic changes in the coming decades likely increase the population's vulnerability to extreme heat substantially. Previous studies have suggested excess deaths could range from 4 to 7 times[46] or even over 18 times[47] higher than current rates in higher emission scenarios. While our results align with a similar magnitude of increase, the estimated excess death cases differ due to variations in methods used to estimate heat-related mortality. Our results highlight that effective safety plans against extreme heat are needed that consider (1) the general risk due to continued greenhouse gas emissions and warming and (2) the dynamic variations due to heat wave occurrence at the regional level.

When compared to the heat risk reports from the RKI, our model predicted more heat-related deaths. These differences can be explained by the variations in the temporal and spatial resolution of the data used in the two models. The RKI reports divided Germany into four regions and used weekly average temperatures as regressors. The use of weekly temperatures is expected to underestimate the variability of the daily temperatures and the impact of the lag effect of heatwaves, thus underestimating the excess death rate[25]. Additionally, the aggregation of temperature data over larger areas is also expected to underestimate the temperature variance across different districts, particularly the heat island effect. For comparison, we also used the average temperature of larger areas as inputs for our model, and the estimated heat-related mortality of our model reached good agreement with the results in the RKI reports (see Supplementary Table 1).

In traditional statistical methods, models typically consist of three variable components: yearly trend, seasonal variation, and temperature response. Our model did not include seasonal variation, yet it achieved similar accuracy in predicting total mortality for hot weeks compared to traditional methods. These results suggest that temperature is the primary driving factor for excess mortality in summer. Therefore, using only temperature without an additional seasonal variable is sufficient to model heat-related mortality.

Previous studies have compared various heat indicators in mortality estimation. The results of our model also show good accordance with previous findings, showing that mean temperature best explains the variation in

summer mortality, as indicated in[48]. Furthermore, we found that daily minimum temperature explains the variation in summer mortality better than daily maximum temperature (see Supplementary Table 5 for details). These observations confirm the findings of previous studies, suggesting that high nighttime temperatures during a heatwave are more fatal to vulnerable populations[49].

This study has several limitations worth acknowledging. First, the model assumed a uniform mortality baseline and temperature-mortality relationship for each age group and gender, regardless of the district. While our results suggested this assumption is acceptable for modelling the heat-related death in Germany, applying a single model to larger areas may be inappropriate. To improve its accuracy, it is necessary to divide the target area based on socio-economic conditions and climate classifications, and tune the model accordingly. Second, although the model demonstrated better accuracy in predictions using mean temperatures at the district level, derived from a dataset with a $1 \times 1$ km resolution, obtaining the precise temperature of every district is challenging. According to the work of Nikolaou et al., a comparison of the dataset with other resources revealed an RMSE of 0.90 °C[33]. These differences can substantially impact the model's accuracy. Furthermore, temperatures can vary largely within a district[50]. These temperature variations lead to varying heat-related risks within a district, which are not accounted for in our model. Third, the climate downscaling model was trained on the ERA5 reanalysis[36] but applied to the climate projection computed with the EC-Earth3 model (see details in the Methods section). In the scope of this study, we did not apply bias correction due to potential additional processing uncertainties[51]. Nevertheless, potential biases in excess death estimation due to the temperature differences between the two data sets should be considered when interpreting the shown results. Fourth, we explored a constant population and base mortality rate for heat-related projections, along with two other demographic scenarios. While the results suggested that an aging population in Germany will increase overall heat-related mortality risk, we did not include scenarios where improvements in the healthcare system could reduce baseline mortality and, consequently, heat-related mortality risk. Moreover, the trained parameters in the model represent the current exposure-response curve, which will likely change with adaptation to heat. Predicting such adaptation changes is very challenging. Nevertheless, our results provide valuable insights for climate policy decisions, emphasizing the importance of proactive measures and long-term planning to mitigate future heat-related health risks effectively.

Many governments and public health authorities are accelerating their efforts to mitigate the impact of heatwaves on public health systems[52,53]. Our model has the potential to assess the effectiveness of such measures. An effective heat prevention plan should result in a notable reduction in registered death cases during a heatwave comparing to the estimation of a model based on historical data. Therefore, we can conclude that such a plan is effective if the registered deaths consistently remain lower than the predicted deaths.

Our model has enabled us to conduct in-depth analyses on the impact of heat on the public health system in Germany. We are able to investigate the effects of heat on specific age groups to identify those most vulnerable. However, characterizing the risk for different age groups is beyond the scope of this paper and will be addressed in future research.

Due to global warming, people are likely to be exposed to extreme heat more often, and public health systems will experience large burdens during the heatwaves. Effective heat prevention plans are becoming crucial in light of this trend. Although our model is trained only on the data from Germany in this study, the model could be applied to study heat-related mortality in other regions. Combined insights can guide national and international climate-driven health policies to address heatwave impacts in the near and long-term future.

## Data availability
All source materials on population and mortality are publicly available[28–31] at the following websites: https://www.destatis.de/and https://www.

regionalstatistik.de/. The climate data from CERRA, ERA5, and DWD are also publicly available[32,34,36] at https://cds.climate.copernicus.eu/and https://www.dwd.de/DE/klimaumwelt/cdc/cdc_node.html. High-resolution climate data is published[33] and is available under a Data Transfer Agreement with Dr. Alexandra Schneider from Helmholtz Munich. Climate projection data is publicly available from various providers (e.g., https://aims2.llnl.gov/search), and the processed data can be provided upon request from the corresponding author due to size constraints. Geographic information data is publicly available (e.g., https://github.com/isellsoap/deutschlandGeoJSON).

One example of the trained parameters of the neural network from 20 ensembles is included in the code as a PyTorch checkpoint file to illustrate the modeling process. The remaining trained parameters are not provided, as the different ensembles were primarily used to analyze the variance within the model and are not essential for understanding or reproducing the main findings. However, they are available upon request from the corresponding author.

## Code availability
The code for the model is publicly available[54] at https://doi.org/10.5281/zenodo.13348002. The interpolation of ERA5 data was performed using the SciPy package (version 1.11.4) in Python. The machine learning model was implemented with the PyTorch (version 2.0.0) and Lightning (version 2.0.3) packages, and it was trained on an NVIDIA Ampere A100 PCIe 40GB GPU.

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

## Acknowledgements
The Federal Ministry of Education and Research (BMBF) supports this study by funding the CLIMADEMIC project (funding code 01LN2210A) within the framework of the Strategy Research for Sustainability (FONA).

## Author contributions
J.W., M.a.d.H. and C.I. conceived the project. J.W. and N.N. collected, processed and validated the data. J.W. and C.I. designed the model. J.W., M.a.d.H and C.I. carried out the investigation. J.W. and C.I. wrote the original draft of the manuscript. N.N. and M.a.d.H. contributed to the revision and result interpretation. C.I. supervised the project.

## Funding

## Competing interests
The authors declare no competing interests.
