## [Peer Review File · Communications Medicine]

Reviewers' comments:

Reviewer #1 (Remarks to the Author):

The authors develop a multi-scale model to estimate daily mortality attributable to heat. They apply this model to assess deaths attributable to heat in Germany at the district level over the last decade and make projections of the heat death burden in that country by the end of the century under three different warming pathways.

This work focuses on a societal problem of widespread importance, the relationship with its explanatory factors not yet clearly understood. It incorporates interesting new features, such as the ability to ingest data with different frequencies. More importantly, their model enables quasi real-time monitoring of deaths during extreme heat events, which can have significant practical implications. However, they only look at a relationship with a few regressors, likely insufficient to describe all the spatial variability of their data. On the other hand, climate change projections lack sociodemographic development, which may result in limited usability of their projections.

Specific comments:

- Title: In my opinion, there should be explicit reference to the studied area in the title.
- L24: The authors refer to 'excess mortality rates' but provide total deaths.
- L26: Please clarify that the provided number refers to cumulative deaths over the analysed period.
- Reference [17] reads incomplete (doi: 10.3238/arztebl.m2022.0202). The authors could use the referred work as a benchmark to compare their results.
- I would add more clarifications in the text to describe what 'multi-scale' means in their modelling context and the potential benefits this feature can bring along. For example, in L83, when the novelty and versatility in modelling are discussed, more clarifications should be included.
- L96: The authors use a temperature cap of 20°C, which may differ across districts. Please assess how this could impact your results.
- Fig. 1: The authors report 48,000 cumulative deaths over the 2014-2023 decade (average: 4,800), but only 3,900 in an exceptionally hot year like 2023. As read in L96, they only consider "the current year until Sep 24", but not many heat-related deaths can be expected afterwards. How do the authors reconcile this evidence?
- Population age is used as a regressor, but no results pay attention to the different population risk, differences in deaths by age groups, or any other aspect relative to age, which is known to be a key determinant of risk.
- L132: I would say 'highlighting the need for preventive measures' rather than 'adaptive measures', given that heat load is already accumulated.

- L148-150: As pointed out before, the assumption that population and death rate are kept constant is a strong assumption that can limit the validity of the projections reported here. This should be extensively discussed.
- Is there a reason to confine Ext. data Fig. 1 and Ext. data Table 1 to the Supplementary Material?
- L317: Please provide more details of the weather data used, e.g., number of stations, coverage.

Reviewer #2 (Remarks to the Author):

This study used a multi-scale machine learning model to calculate heat-related excess mortality rates at the district level and visualized local health risks during a selected heatwave. But I am greatly concerned about the model established for predicting the mortality. As the progress of health outcome is complex and multifactorial, it is important to consider all the available factors, such as dietary habit, age, socioeconomic status, history of disease, influenza, and environmental factors (air pollution and temperature), when predicting the death outcome. Only using temperature is totally insufficient and also unreasonable to predict the death outcome, especially when the authors used this model for projecting death outcome in future (such as the year of 2010). In addition, for the projection of future heat-related mortality, the relevant methodology is unclear, such as how do you define the “heat” and “heatwave”, and how heat-related mortality were calculated. My other comments for the introduction and discussion sections are as follows:

1. In the main section, It is crucial to focus on describing the future impact of global warming and emphasize the significance of this study rather than describing changes in human civilization and industrial transformation.
2. For temperature data, it is unclear why these data were from three different sources? Analyzing data from different sources can lead to uncertainty in research result.
3. In Discussion section, The findings of this study are compared with reports from RKI. However, the reliability of the RKI report as a benchmark or just a precise comparison remains uncertain. Thus, the discussion is limited in proposing comparisons between the results of this paper and others.

Reviewers' comments:

Reviewer #1 (Remarks to the Author):

The authors develop a multi-scale model to estimate daily mortality attributable to heat. They apply this model to assess deaths attributable to heat in Germany at the district level over the last decade and make projections of the heat death burden in that country by the end of the century under three different warming pathways.

This work focuses on a societal problem of widespread importance, the relationship with its explanatory factors not yet clearly understood. It incorporates interesting new features, such as the ability to ingest data with different frequencies. More importantly, their model enables quasi real-time monitoring of deaths during extreme heat events, which can have significant practical implications. However, they only look at a relationship with a few regressors, likely insufficient to describe all the spatial variability of their data. On the other hand, climate change projections lack sociodemographic development, which may result in limited usability of their projections.

Specific comments:

- Title: In my opinion, there should be explicit reference to the studied area in the title.
- L24: The authors refer to 'excess mortality rates' but provide total deaths.
- L26: Please clarify that the provided number refers to cumulative deaths over the analysed period.
- Reference [17] reads incomplete (doi: 10.3238/arztebl.m2022.0202). The authors could use the referred work as a benchmark to compare their results.
- I would add more clarifications in the text to describe what 'multi-scale' means in their modelling context and the potential benefits this feature can bring along. For example, in L83, when the novelty and versatility in modelling are discussed, more clarifications should be included.
- L96: The authors use a temperature cap of 20°C, which may differ across districts. Please assess how this could impact your results.
- Fig. 1: The authors report 48,000 cumulative deaths over the 2014-2023 decade (average: 4,800), but only 3,900 in an exceptionally hot year like 2023. As read in L96, they only consider "the current year until Sep 24", but not many heat-related deaths can be expected afterwards. How do the authors reconcile this evidence?
- Population age is used as a regressor, but no results pay attention to the different population risk, differences in deaths by age groups, or any other aspect relative to age, which is known to be a key determinant of risk.
- L132: I would say 'highlighting the need for preventive measures' rather than 'adaptive measures', given that heat load is already accumulated.
- L148-150: As pointed out before, the assumption that population and death rate are kept constant is a strong assumption that can limit the validity of the projections reported here. This should be extensively discussed.
- Is there a reason to confine Ext. data Fig. 1 and Ext. data Table 1 to the Supplementary Material?
- L317: Please provide more details of the weather data used, e.g., number of stations, coverage.

Reviewer #2 (Remarks to the Author):

This study used a multi-scale machine learning model to calculate heat-related excess mortality rates at the district level and visualized local health risks during a selected heatwave. But I am greatly concerned about the model established for predicting the mortality. As the progress of health outcome is complex and multifactorial, it is important to consider all the available factors, such as dietary habit, age, socioeconomic status, history of disease, influenza, and environmental factors (air pollution and temperature), when predicting the death outcome. Only using temperature is totally insufficient and also unreasonable to predict the death outcome, especially when the authors used this model for projecting death outcome in future (such as the year of 2010). In addition, for the projection of future heat-related mortality, the relevant methodology is unclear, such as how do you define the "heat" and "heatwave", and how heat-related mortality were calculated. My other comments for the introduction and discussion sections are as follows:

1. In the main section, It is crucial to focus on describing the future impact of global warming and emphasize the significance of this study rather than describing changes in human civilization and industrial transformation.
2. For temperature data, it is unclear why these data were from three different sources? Analyzing data from different sources can lead to uncertainty in research result.
3. In Discussion section, The findings of this study are compared with reports from RKI. However, the reliability of the RKI report as a benchmark or just a precise comparison remains uncertain. Thus, the discussion is limited in proposing comparisons between the results of this paper and others.

Authors' response

We thank the reviewers for the constructive comments to help us to improve our manuscript. The original comments are listed below followed by our response in blue and manuscript adjustments in green.

Response to reviewer #1

The authors develop a multi-scale model to estimate daily mortality attributable to heat. They apply this model to assess deaths attributable to heat in Germany at the district level over the last decade and make projections of the heat death burden in that country by the end of the century under three different warming pathways.

This work focuses on a societal problem of widespread importance, the relationship with its explanatory factors not yet clearly understood. It incorporates interesting new features, such as the ability to ingest data with different frequencies. More importantly, their model enables quasi real-time monitoring of deaths during extreme heat events, which can have significant practical implications. However, they only look at a relationship with a few regressors, likely insufficient to describe all the spatial variability of their data. On the other hand, climate change projections lack sociodemographic development, which may result in limited usability of their projections.

Thank you for the thorough review. Your comments are extremely helpful for the improvement of our manuscript.

Regarding your concern of the manuscript, it is typical only to use temperature as the regressor in studies of heat-related mortality [1, 2, 3, 4]. The other factors mainly affect the baseline mortality rate and we discussed the impact of baseline mortality rate selection on model accuracy in Supplementary Section 2.

With the climate projection for different shared socioeconomic pathways (SSPs), we also considered different sociodemographic developments. However, we did not consider the impact of sociodemographic developments on factors other than climate change. We will address this problem again when replying to your further comments.

Specific comments:

- Title: In my opinion, there should be explicit reference to the studied area in the title.

We include Germany in our title and now the title reads

High-Resolution Modeling and Projection of Heat-Related Mortality in Germany under Climate Change

- L24: The authors refer to 'excess mortality rates' but provide total deaths.

We rewrote the text to:

... we **estimated heat-related excess mortality** at the district level ...

- L26: Please clarify that the provided number refers to cumulative deaths over the analysed period.

We rewrote the text to:

... we estimated **a total of 48,000 heat-related deaths** in Germany during the last decade ...

- Reference [17] reads incomplete (doi: 10.3238/arztebl.m2022.0202). The authors could use the referred work as a benchmark to compare their results.

Thank you for pointing out the error. We completed the mentioned citation. The referred work was used as a benchmark in line 197-201 under another reference number [31], as the data for a direct comparison is available in that publication.

- I would add more clarifications in the text to describe what 'multi-scale' means in their modelling context and the potential benefits this feature can bring along. For example, in L83, when the novelty and versatility in modelling are discussed, more clarifications should be included.

We agree that more clarification is needed for explaining the term 'multi-scale' and added the following text to line 91-94:

...**The term 'multi-scale' refers to the utilization of data with varying temporal resolutions, ranging from daily to weekly, and data with different spatial resolutions, from 1x1 km grids to statewide. This multi-scale approach enables a risk assessment with higher temporal and spatial resolutions comparing to other models...**

- L96: The authors use a temperature cap of 20°C, which may differ across districts. Please assess how this could impact your results.

Thank you for the suggestion. In our method, we assumed that the impact of the temperature is the same across Germany so that we have enough data for training the model. Therefore, the differences across districts are not applicable in our model. Nevertheless, the choice of the cap temperature has a significant impact on the estimated excess mortality. Also, the cap temperature of 20 °C is used to be consistent with other publications, especially with the one from RKI [4]. We added a section in the supplementary materials addressing the issue and added the following sentence to the main text line 110-111.

...(The impact of the cap temperature is discussed in Supplementary Section 3)...

- Fig. 1: The authors report 48,000 cumulative deaths over the 2014-2023 decade (average: 4,800), but only 3,900 in an exceptionally hot year like 2023. As read in L96, they only consider “the current year until Sep 24”, but not many heat-related deaths can be expected afterwards. How do the authors reconcile this evidence?

Thank you for this very interesting comment. Indeed, Germany experienced the highest yearly average temperature in 2023, characterised by exceptional high average temperatures in autumn[5]. Although the temperatures in autumn of 2023 were higher than in other years, they still remained below the cap temperature used for calculating heat-related deaths almost all the time. The summer temperatures in 2023, in contrast, were not as high as in 2018-19 and 2022[5]. The intensity and geographic distribution of the heatwaves were also different, which explains the comparably low estimated heat-related mortality with regard to the hotter summers of 2018/19/22.

To assert this discussion, we have updated the manuscript with the now available and complete weather data of 2023 and the new selection of weather stations. The estimated heat-related mortality changed from 5500 in 2022 and 3900 in 2023 to 5600 in 2022 and 4000 in 2023. Both changes are within the range of the uncertainty of the model. These results confirm our arguments.

- Population age is used as a regressor, but no results pay attention to the different population risk, differences in deaths by age groups, or any other aspect relative to age, which is known to be a key determinant of risk.

Thank you for raising this issue. The age is indeed a key determinant of the risk, which was also considered in our model. The main reason that we exclude the results and discussions about the impact of the age on the risk is that we want to focus on introducing the new modelling tools and show what we can do with the new model. An extensive discussion of the age induced risk would go beyond the scope of this paper, as we mentioned in line 302-306. We strongly agree that age is an important factor and we are working on another paper to discuss it separately.

- L132: I would say ‘highlighting the need for preventive measures’ rather than ‘adaptive measures’, given that heat load is already accumulated.

We agree with you and added preventive to the manuscript. We also kept the word adaptive because we think it is important to have correspondent adaptive measures in health care system. Now the manuscript reads
...highlighting the need for **preventive** and adaptive measures...

- L148-150: As pointed out before, the assumption that population and death rate are kept constant is a strong assumption that can limit the validity of the projections reported here. This should be extensively discussed.

Thank you for this important suggestion. We added discussion about this limitation in line 284-294.

...**Fourth, we assumed a constant population and base mortality rate for heat-related projections, acknowledging that socioeconomic factors will significantly impact heat-related deaths. Aging population in Germany will increase heat-related mortality risk, while improvements in the health care system will reduce baseline mortality and also reduce heat-related mortality risk. Moreover, the trained parameter in the model represent current exposure-response curve and will also change with the adaption to heat. However, to the best of our knowledge, we could neither find a reliable projection for district-level population and age structure nor a projection of age-specific death rates by the end of the century. A prognosis of the change of the adaption is also difficult. Thus, we decided to assume a constant population and death rate to avoid introducing further uncertainties...**

- Is there a reason to confine Ext. data Fig. 1 and Ext. data Table 1 to the Supplementary Material?

We chose to confine Ext. data Fig. 1 and Table 1 to the supplementary as they are not the key result of the paper. In our opinion, they are mainly a validation of our method.

- L317: Please provide more details of the weather data used, e.g., number of stations, coverage.

Thank you for the suggestion. We added the number of the stations used in the paper and a map illustrating the coverage of the stations:

... **We gathered information from 537 weather stations. These stations have been measuring the average daily air temperature 2 meters above the ground since 2010 and still do so until at least 2022. The location of the stations is illustrated in Extended Data Figure 4...**

Response to reviewer #2

This study used a multi-scale machine learning model to calculate heat-related excess mortality rates at the district level and visualized local health risks during a selected heatwave. But I am greatly concerned about the model established for predicting the mortality. As the progress of health outcome is complex and multifactorial, it is important to consider all the available factors, such as dietary habit, age, socioeconomic status, history of disease, influenza, and environmental factors (air pollution and temperature), when predicting the death outcome. Only using temperature is totally insufficient and also unreasonable to predict the death outcome, especially when the authors used this model for projecting death outcome in future (such as the

year of 2010). In addition, for the projection of future heat-related mortality, the relevant methodology is unclear, such as how do you define the “heat” and “heatwave”, and how heat-related mortality were calculated. My other comments for the introduction and discussion sections are as follows:

Thank you very much for the thorough review. We want to clarify that our model not only utilizes temperature data but also incorporates population and base mortality rate information. Further, the function of our model is not to predict mortality of certain individual, but rather to estimate how the population level mortality risk changes with temperature, i.e. the health risk due to heat.

We acknowledge the impact of various factors, such as dietary habits, age, socioeconomic status, history of disease, influenza, and environmental factors (such as air pollution and temperature) on mortality at the individual level. Indeed, influenza has a very strong impact on the performance of our model. In Section 4, we discussed that our model performs worse for colder months, partially due to influenza waves. However, the impacts of most factors are averaged out at the population level and were already accounted for by the proper selection of base mortality rate. In the supplementary materials, we extensively discussed how the selection of the mortality baseline affected the accuracy of the model. In line 182-186 we mentioned that we reached a RMSE error of 90 for warm days, which is close to the theoretically lower bound of 45-60 based on Poisson distribution. This also confirms that it is acceptable to neglect other impact factors when estimating heat-related mortality, as in previous works[1, 2, 3, 4]. We recognized that our manuscript lacks necessary discussions regarding the assumption of constant population and mortality rates, and we have addressed this limitation in the discussion section.

...Fourth, we assumed a constant population and base mortality rate for heat-related projections, acknowledging that socioeconomic factors will significantly impact heat-related deaths. Aging population in Germany will increase heat-related mortality risk, while improvements in the health care system will reduce baseline mortality and also reduce heat-related mortality risk. Moreover, the trained parameter in the model represent current exposure-response curve and will also change with the adaption to heat. However, to the best of our knowledge, we could neither find a reliable projection for district-level population and age structure nor a projection of age-specific death rates by the end of the century. A prognosis of the change of the adaption is also difficult. Thus, we decided to assume a constant population and death rate to avoid introducing further uncertainties....

In line 224-226 we mentioned that we did not explicitly define a heatwave. This is because the definition of a heatwave varies significantly across different studies. In contrast, our model does not require an explicit definition of a heatwave. Instead, it captures heatwave characteristics through temperature data and mortality data. We acknowledged that we did not provide enough information about this issue and we added the following text to the manuscript in line 219-221:

...The contribution of heatwaves to the total heat-related mortality has not been extensively studied. Xu et al. compared different definition of heatwaves in a review paper and showed that defining a heatwave across large areas is difficult due to the diversities in population acclimatization and adaptation. Pascal et al. suggested...In this paper, we did not explicitly define a heatwave, instead, our model captures heatwave characteristics through temperature and mortality data...the top 10 days with the highest heat-related deaths each year contributed to 50-80% of the total heat-related mortality of the corresponding year in Germany. Only 7% of the days are disconnected, which signifies the impact of consecutive hot days, i.e. heatwaves....

In line 109-111 we described how heat-related mortality was calculated. In response to your comment, we have added information in the supplementary materials on how different temperature caps affect the estimation, and added the following sentence to the main text:

...(The impact of the cap temperature is discussed in Supplementary Section 3)...

1. In the main section, It is crucial to focus on describing the future impact of global warming and emphasize the significance of this study rather than describing changes in human civilization and industrial transformation.

We agree with you and added following text to the manuscript in line 62-73:

As a consequence of the rising heat-related burden, temperature has become an essential global monitoring variable to assess current and future health risks. Currently, we are on track for 2.5-2.9°C of global warming under current climate pledges this century, which will further intensify the described climate impacts and, thus, also the burden on our health systems. Several nations are in the process of shaping and implementing adaptation and safety plans against extreme heat. Although many studies have assessed the impact of heat on mortality, most either provide risk estimation for large regions or only focus on cities. Combined risk estimation and projection across different spatial and temporal scales is currently still missing, making it challenging for local governments to make targeted preventive and adaptive plans. Accurate and high-resolution estimates of the heat-related stress for public health systems are crucial for such purpose.

2. For temperature data, it is unclear why these data were from three different sources? Analyzing data from different sources can lead to uncertainty in research result.

Thank you for raising the issue. The data from Helmholtz Munich and Copernicus are used separately to train the model. Our target was to demonstrate the impact of the resolution of temperature data on model accuracy, and this was extensively discussed in Supplementary Section 2. As mentioned in line 343-345, the data from Helmholtz Munich is not in real-time. For real-time risk analysis, we trained a model to map the data from weather stations to the district level. These are the reasons

why we are using three different temperature datasets.

3. In Discussion section, The findings of this study are compared with reports from RKI. However, the reliability of the RKI report as a benchmark or just a precise comparison remains uncertain. Thus, the discussion is limited in proposing comparisons between the results of this paper and others.

Since there are no exact records for heat-related deaths available in Germany, which could serve as ground truth, we have to rely on model-based estimates. Similarly, several countries and studies are focusing on the usage of statistical tools to estimate heat-related mortality based on temperature data[1, 2, 3, 4, 6]. Yet, we argue that we can increase our confidence in said estimates, especially when completely different models (e.g., the ML model of this study and the statistical model behind the current RKI reports) reach agreement in terms of their predictions. We acknowledged that our text and supplementary are not clear, therefore, we modified the text in line 261-264 and added the RKI estimation to the Supplementary Table 7.

...particularly the heat island effect. **For comparison, we also used the average temperature of larger areas as inputs for our model, and the estimated heat-related mortality of our model reached good agreement with the results in the RKI reports.**

References

- [1] Joan Ballester, Marcos Quijal-Zamorano, Raúl Fernando Méndez Turrubiates, Ferran Pegenaute, François R Herrmann, Jean Marie Robine, Xavier Basagaña, Cathryn Tonne, Josep M Antó, and Hicham Achebak. Heat-related mortality in europe during the summer of 2022. *Nature Medicine*, pages 1–10, 2023.
- [2] Ana Maria Vicedo-Cabrera, N Scovronick, Francesco Sera, Dominic Royé, Rochelle Schneider, Aurelio Tobias, Christofer Astrom, Y Guo, Y Honda, DM Hondula, et al. The burden of heat-related mortality attributable to recent human-induced climate change. *Nature climate change*, 11(6):492–500, 2021.
- [3] Samuel Lüthi, Christopher Fairless, Erich M Fischer, Noah Scovronick, Ben Armstrong, Micheline De Sousa Zantotti Stagliorio Coelho, Yue Leon Guo, Yuming Guo, Yasushi Honda, Veronika Huber, et al. Rapid increase in the risk of heat-related mortality. *Nature communications*, 14(1):4894, 2023.
- [4] Matthias an der Heiden, Christoph Winklmayr, Sabine Buchien, Marion Schranz, RKI-Geschäftsstelle für Klimawandel & Gesundheit, Michael Diercke, and Viviane Bremer. Wochenbericht zur hitzebedingten mortalität kw 38/2023 vom 05.10.2023. 2023.
- [5] Time series and trends for the parameters temperature, precipitation, sunshine duration and various climate indices.
- [6] Claudia Winklmayr, Stefan Muthers, Hildegard Niemann, Hans-Guido Mücke, and Matthias an der Heiden. Heat-related mortality in germany from 1992 to 2021. *Deutsches Ärzteblatt International*, 119(26):451.

Reviewers' comments:

Reviewer #1 (Remarks to the Author):

Thank you for your detailed rebuttal letter and the revised version of your manuscript, now titled "High-Resolution Modeling and Projection of Heat-Related Mortality in Germany under Climate Change." I appreciate the effort you have put into addressing the comments and suggestions provided in my initial review. Below, I provide my assessment of the revisions and further comments.

I am very satisfied with the clarifications and amendments, which have improved the manuscript. The revisions have addressed the majority of the concerns raised and have enhanced the clarity and rigour of the study.

While the manuscript is substantially improved, one of my main concerns persists: future death rates are assumed to be held constant, and future population developments are not included, which gives a biased picture of the future number of deaths attributed to heat in the studied area. This would be a minor issue if the projection exercise was deemed an accessory or complementary experiment. However, this does not seem to be the case. Death projections are a central focus of the paper (as indicated by the term 'projection' in the title).

I understand that it may be too demanding to repeat the entire analysis with these factors taken into account, and it is true that this limitation is acknowledged in the discussion. However, for the sake of clarity and to avoid confusing the reader, I suggest stating clearly from the start that projections are subject to 'static' sociodemographic developments. I leave it to the author to decide whether this should be done in the abstract or in the introduction.

In conclusion, I believe the manuscript is now much closer to being suitable for publication, especially after the above issue has been addressed.

Sincerely,

Reviewer #2 (Remarks to the Author):

The authors have addressed most of my comments. But here, I still have several concerns, particularly for the methodology.

1. In the first equation of “6.2 Machine learning...predictions”, $f_{age, sex}(T_t)$ should be the mortality risk of high temperature, but in the following equation, this term seems to be the function of temperature. More explanation is required.

2. Accuracy of the prediction model needs to be greatly improved:

1) The authors reported the RMSE close to 100, but it is not clear for me why this model has good performance when RMSE close to 100.

2) More indicator for assessing the model performance (such as R-squared) is suggested.

3) How did you select the training and validation data, randomly or other way?

4) The data are at district level. Did you establish the model at district level and then combine the results (such as RMSE) together? If so, more information is required on how did you combine results at district level.

5) Definition for the heat is required. And how do you obtain the minimum mortality temperature?

6) Lines 383-384, temperature could only explain part of season trends of mortality. Other time-varying variables such as air pollutants, humidity could be also associated with risk of mortality.

3. The results on the model performance of different temperature measures (minimum, mean, maximum) are interesting. I would suggest move these results to the main texts from the supplementary materials.

4. The evidence and the prediction model may be used to improve the current heat-health warning system. But I do not think it is good idea to project the for the much long-term mortality risk (such as at 2099). There is large uncertainty in projected temperature in fat future, and

other uncertainties (such as those from temperature-mortality association, urbanization, adaptation to future heat) are totally not considered.

Reviewers' comments:

Reviewer #1 (Remarks to the Author):

Thank you for your detailed rebuttal letter and the revised version of your manuscript, now titled "High-Resolution Modeling and Projection of Heat-Related Mortality in Germany under Climate Change." I appreciate the effort you have put into addressing the comments and suggestions provided in my initial review. Below, I provide my assessment of the revisions and further comments.

I am very satisfied with the clarifications and amendments, which have improved the manuscript. The revisions have addressed the majority of the concerns raised and have enhanced the clarity and rigour of the study.

While the manuscript is substantially improved, one of my main concerns persists: future death rates are assumed to be held constant, and future population developments are not included, which gives a biased picture of the future number of deaths attributed to heat in the studied area. This would be a minor issue if the projection exercise was deemed an accessory or complementary experiment. However, this does not seem to be the case. Death projections are a central focus of the paper (as indicated by the term 'projection' in the title).

I understand that it may be too demanding to repeat the entire analysis with these factors taken into account, and it is true that this limitation is acknowledged in the discussion. However, for the sake of clarity and to avoid confusing the reader, I suggest stating clearly from the start that projections are subject to 'static' sociodemographic developments. I leave it to the author to decide whether this should be done in the abstract or in the introduction.

In conclusion, I believe the manuscript is now much closer to being suitable for publication, especially after the above issue has been addressed.

Sincerely,

Reviewer #2 (Remarks to the Author):

The authors have addressed most of my comments. But here, I still have several concerns, particularly for the methodology.

1. In the first equation of "6.2 Machine learning... predictions", $f_{age, sex}(T_t)$ should be the mortality risk of high temperature, but in the following equation, this term seems to be the function of temperature. More explanation is required.

2. Accuracy of the prediction model needs to be greatly improved:

1) The authors reported the RMSE close to 100, but it is not clear for me why this model has good performance when RMSE close to 100.

2) More indicator for assessing the model performance (such as R-squared) is suggested.

3) How did you select the training and validation data, randomly or other way?

4) The data are at district level. Did you establish the model at district level and then combine the results (such as RMSE) together? If so, more information is required on how did you combine results at district level.

5) Definition for the heat is required. And how do you obtain the minimum mortality temperature?

6) Lines 383-384, temperature could only explain part of season trends of mortality. Other time-varying variables such as air pollutants, humidity could be also associated with risk of mortality.

3. The results on the model performance of different temperature measures (minimum, mean, maximum) are interesting. I would suggest move these results to the main texts from the supplementary materials.

4. The evidence and the prediction model may be used to improve the current heat-health warning system. But I do not think it is good idea to project the for the much long-term mortality risk (such as at 2099). There is large uncertainty in projected temperature in far future, and other uncertainties (such as those from temperature-mortality association, urbanization, adaptation to future heat) are totally not considered.

Authors' response

We thank the reviewers for the comments in the second round to help us to further improve our manuscript. The original comments are listed below followed by our response in blue and manuscript adjustments in green. In summary, we have made several modification to the main text and the supplementary material to further clarify and discuss the results of our study. In addition, we have performed a new set of simulations to account for the previously assumed static future demographic development. With these new simulations, we can examine, the effect of future population aging wrt. the already increasing heat-related health risks. Please find all details below and in the new version of the manuscript.

Response to reviewer #1

Thank you for your detailed rebuttal letter and the revised version of your manuscript, now titled "High-Resolution Modeling and Projection of Heat-Related Mortality in Germany under Climate Change." I appreciate the effort you have put into addressing the comments and suggestions provided in my initial review. Below, I provide my assessment of the revisions and further comments.

I am very satisfied with the clarifications and amendments, which have improved the manuscript. The revisions have addressed the majority of the concerns raised and have enhanced the clarity and rigour of the study.

We are also grateful for your help to improve our manuscript.

While the manuscript is substantially improved, one of my main concerns persists: future death rates are assumed to be held constant, and future population developments are not included, which gives a biased picture of the future number of deaths attributed to heat in the studied area. This would be a minor issue if the projection exercise was deemed an accessory or complementary experiment. However, this does not seem to be the case. Death projections are a central focus of the paper (as indicated by the term 'projection' in the title).

I understand that it may be too demanding to repeat the entire analysis with these factors taken into account, and it is true that this limitation is acknowledged in the discussion. However, for the sake of clarity and to avoid confusing the reader, I suggest stating clearly from the start that projections are subject to 'static' sociodemographic developments. I leave it to the author to decide whether this should be done in the abstract or in the introduction.

Thank you for pointing out this issue. We added the clarification in the abstract as you suggested.

...Combining our model with shared socio-economic pathways (SSPs) of future climate change provides evidence that heat-related mortality in Germany could further increase by a factor of 2.5 (SSP245) to 9 (SSP370) without adaptation to extreme heat under static sociodemographic developments assumption...

In conclusion, I believe the manuscript is now much closer to being suitable for publication, especially after the above issue has been addressed.

Sincerely,

Response to reviewer #2

The authors have addressed most of my comments. But here, I still have several concerns, particularly for the methodology.

Thank you very much for your time and your helpful remarks. We have made further adjustments based on your comments to clarify the methodology in the manuscript.

1. In the first equation of "6.2 Machine learning... predictions", $f_{age, sex}(T_t)$ should be the mortality risk of high temperature, but in the following equation, this term seems to be the function of temperature. More explanation is required.

We carefully checked the equation and we believe that the equation is correct. The notation $f_{age, sex}(T_t)$ is indeed a function of temperature and the output of the function is the mortality risk. In order to make it clearer, we added following text to the manuscript to describe function f .

We used temperature data (T_t) of each district at time t as input for our model to obtain the result $f_{age, sex}(T_t)$. **The function f is used to describe the temperature dependent variability of temperature-related risks.** The model accounted for the lag effects...

2. Accuracy of the prediction model needs to be greatly improved:

1) The authors reported the RMSE close to 100, but it is not clear for me why this model has good performance when RMSE close to 100.

In previous studies, Poisson or quasi-Poisson regression models are most often used to describe the relationship between temperature and mortality [1, 2, 3]. In a Poisson distribution, the standard deviation is equal to the mean of the distribution. Therefore, with a perfect Poisson distribution assumption, the RMSE is expected to be same as the square root of the average, which is around 50-60. Consequently, a daily RMSE of 80-90 is a good result compared to the theoretical lower bound and

also in comparison to other experimental setups in this study. We found the weekly prediction for Germany in another study [4], and used it as a comparison to our study. The comparison demonstrated that our method also captures the impact of heat on mortality. We strongly agree that RMSE alone is not enough to demonstrate the performance of the model, and we have added the R-value according to the next comment.

2) More indicator for assessing the model performance (such as R-squared) is suggested.

Thank you very much for the suggestion. Adding the R-value helps us demonstrate the results more effectively. We have added the R-squared value to the main manuscript on line 189 and line 207, as well as in the supplementary material. Additionally, we included a scatter plot in the supplementary material to show the differences between the death predictions of our model and the registered death cases, compared to the model used in the previous publication[4].

Extended Data Table 1 demonstrates that our model had an accurate estimation of the daily mortality cases in Germany for warmer days (national-wide mean temperature above 20°C. Root-Mean-Square Error (RMSE): 91.4 for training data, 83.9 for validation data. **Coefficient of determination(R^2): 0.7887 for training data, 0.8272 for validation data.**). For comparison, the theoretical lower bound of the RMSE falls within the range of 45 to 60 according to the Poisson distribution **and the upper bound of the R^2 is around 0.94**. However, when considering all days in a year, the predictions are less accurate (RMSE: 171.1 for training data, 199.6 for validation data. R^2 : **0.5961 for training data, 0.4884 for validation data.**)...

... For weeks with an average temperature above 20°C, our model demonstrated performance similar to that of the statistical model used in RKI reports (weekly mortality RMSE from 2011 to 2020, except for 2013: 362.1 for our model, 364.9 for the GAM model used in RKI reports, **weekly R^2 : 0.8997 for our model, 0.8982 for the GAM model used in RKI reports.**)

3) How did you select the training and validation data, randomly or other way?

Thank you for the comment, we have clarified the selection of the training/validation data sets in Line 426.

...maintaining an 80:20 ratio **with data from 2011-2018 used as training data and data from 2019-2020 as validation data.**

4) The data are at district level. Did you establish the model at district level and then combine the results (such as RMSE) together? If so, more information is required on how did you combine results at district level.

The initial data sources are provided at different spatial scales. While the temperature data are available at district level (1x1 km grids in the raw format), the target mortality data are provided at varying levels from district level to nationwide. The neural network model itself does not contain any specific information about the districts. The information at district level is always included in the input (temperature, registered mortality and population). The predicted risk factor with such input are then at district level, and we used it to estimate the mortality in each district. Then we aggregated the data according to the scales of the target, and used Poisson loss to update the parameters. The whole process was described in lines 411-424.

5) Definition for the heat is required. And how do you obtain the minimum mortality temperature?

The target of our model is to predict the all-cause mortality based on the input (temperature, population and past mortality data). Therefore a definition of heat is not required for the model. However, we defined heat-related mortality when interpreting the output of the model (lines 310-312). The minimum mortality temperature can be obtained by finding the maximum of extended figure 3, which is around 18°C. For our analysis, however, we selected 20°C as baseline temperature for the definition of heat-related mortality to be consistent with the current reports from Robert Koch Institute.

6) Lines 383-384, temperature could only explain part of season trends of mortality. Other time-varying variables such as air pollutants, humidity could be also associated with risk of mortality.

Thank you for pointing out the missing explanation of the results of our model. Indeed, we acknowledge that the factors you mention pose additional health threats, especially on an individual level. For our predictive modelling, however, we show that the predictive power of temperature alone is a sufficient proxy for heat-related excess mortality in summer. This, of course, also changes during winter months, when other non-temperature related drivers become dominant. We added another paragraph explaining this in lines 280-286:

In traditional statistical methods, models typically consist of three variable components: yearly trend, seasonal variation, and temperature response. Our model did not include seasonal variation, yet it achieved similar accuracy in predicting total mortality for hot weeks compared to traditional methods. These results suggest that temperature is the primary driving factor for excess mortality in summer. Therefore, using only temperature without an additional seasonal variable is sufficient to model heat-related mortality.

3. The results on the model performance of different temperature measures (minimum, mean, maximum) are interesting. I would suggest move these results to the main texts from the supplementary materials.

Thank you for the suggestion. We have added a paragraph in the main text discussing this point and kept the tables in the supplementary text (see lines 287-294).

Previous studies have compared various heat indicators in mortality estimation. The results of our model also show good accordance with previous findings. We observed that mean temperature best explains the variation in summer mortality, as indicated in [5]. Furthermore, we found that daily minimum temperature explains the variation in summer mortality better than daily maximum temperature (see Supplementary Section 2 for details). These observations confirm the findings of previous studies, suggesting that high nighttime temperatures during a heatwave are more fatal to vulnerable populations[6].

4. The evidence and the prediction model may be used to improve the current heat-health warning system. But I do not think it is good idea to project the for the much long-term mortality risk (such as at 2099). There is large uncertainty in projected temperature in fat future, and other uncertainties (such as those from temperature-mortality association, urbanization, adaptation to future heat) are totally not considered.

We agree that an exact calculation of long-term mortality risk is impossible, even when considering all factors. Yet, both the short-term forecasts as well as the long-term projections have their own purpose and should be discussed together. Moving from short-term forecasts to long-term projections naturally increases all associated uncertainties substantially. In this context, the different climate scenarios provide an invaluable estimate of potential "what ifs". While scenario-based projections may differ significantly from the actual future (and also between each other), they provide valuable insights into how much climate change may contribute to increased mortality risk in summer, without accounting for other factors. This information is useful for long-term planning and underpins the importance of political decision-making.

Your remark on missing factors, especially demographic drivers, is more than justified. To account for these factors, we obtained future scenarios on demographic changes in Germany and included them in our model approach. The scenarios are based on varying birth rate, life expectancy, and net migration. We picked scenarios that envelope potential future developments in a low and high population aging scenario. In combination with the previous results, we can derive an additional demographic-driven risk factor for our scenario simulations. The results are shown is a new figure (see below) and discussed in the new version of the manuscript. In particular, it is noteworthy that these results indicate that the risk introduced by demographic changes is almost independent of the impact of climate change. In other words, these new results provide further evidence that an aging population, as it is expected in Germany, is at particular risk, even with only moderately rising temperatures.

Figure 1: Left: Population structure in 2070 for the low aging scenario (top panel, slight increase in life expectancy and low net migration, variant 13) and high aging scenario (bottom panel, sharp increase in life expectancy and high net migration, variant 20). The grey bar represents the population in 2021. Right: Relative changes in population-level heat-related mortality risk compared to the static population structure in 2021 as in Fig. 3 of the main text.

Changes in the abstract:

...Combining our model with shared socio-economic pathways (SSPs) of future climate change provides evidence that heat-related mortality in Germany could further increase by a factor of 2.5 (SSP245) to 9 (SSP370) without adaptation to extreme heat **under static sociodemographic developments assumption...**

Changes in the Discussion chapter:

We investigated this effect by utilizing two different non-static demographic projections until the year 2070, simulating a range of future population aging scenarios that are driven by varying birth rate, life expectancy, and net migration (Extended Data Figure 5). Independent of the climate scenario, we observe a steadily increasing additional risk factor until at least 2050 in the range of 40 to 60 % compared to the static population assumption. These results suggest that the expected demographic changes in the coming decades likely increase the population's vulnerability to extreme heat substantially.

Changes in the Discussion chapter about limitations:

Fourth, we explored a constant population and base mortality rate for heat-related projections, along with two other demographic scenarios. While the results suggested that an aging population in Germany will increase overall heat-related mortality risk, we did not include scenarios where improvements in the healthcare system could reduce baseline mortality and, consequently, heat-related mortality risk. Moreover, the trained parameters in the model represent the current exposure-response curve, which will likely change with adaptation to heat. Predicting such adaptation changes is very challenging. Nevertheless, our results provide valuable insights for climate policy decisions, emphasizing the importance of proactive measures and long-term planning to mitigate future heat-related health risks effectively.

References

- [1] Antonio Gasparrini, Ben Armstrong, and Mike G Kenward. Distributed lag non-linear models. *Statistics in medicine*, 29(21):2224–2234, 2010.
- [2] Joan Ballester, Marcos Quijal-Zamorano, Raúl Fernando Méndez Turrubiates, Ferran Pegenaute, François R Herrmann, Jean Marie Robine, Xavier Basagaña, Cathryn Tonne, Josep M Antó, and Hicham Achebak. Heat-related mortality in europe during the summer of 2022. *Nature Medicine*, pages 1–10, 2023.
- [3] Ana Maria Vicedo-Cabrera, N Scovronick, Francesco Sera, Dominic Royé, Rochelle Schneider, Aurelio Tobias, Christofer Astrom, Y Guo, Y Honda, DM Hondula, et al. The burden of heat-related mortality attributable to recent human-induced climate change. *Nature climate change*, 11(6):492–500, 2021.
- [4] Matthias an der Heiden, Christoph Winklmayr, Sabine Buchien, Marion Schranz, RKI-Geschäftsstelle für Klimawandel & Gesundheit, Michael Diercke, and Viviane Bremer. Wochenbericht zur hitzebedingten mortalität kw 38/2023 vom 05.10.2023. 2023.
- [5] M an der Heiden, S Muthers, H Niemann, U Buchholz, L Grabenhenrich, and A Matzarakis. Schätzung hitzebedingter todesfälle in deutschland zwischen 2001 und 2015. *bundesgesundheitsbl.* 62: 571–579, 2019.
- [6] Karine Laaidi, Abdelkrim Zeghnoun, Bénédicte Dousset, Philippe Bretin, Stéphanie Vandentorren, Emmanuel Giraudet, and Pascal Beaudeau. The impact of heat islands on mortality in paris during the august 2003 heat wave. *Environmental health perspectives*, 120(2):254–259, 2012.

REVIEWERS' COMMENTS:

Reviewer #2 (Remarks to the Author):

I suggest the current manuscript is suitable for publication.